

# Interaction effects in a multi-channel Fabry-Pérot interferometer in the Aharonov-Bohm regime

Dario Ferraro[1,2*] and Eugene Sukhorukov[1]

**1** Département de Physique Théorique, Université de Genève, 24 quai Ernest Ansermet, CH-1211 Geneva, Switzerland
**2** Aix Marseille Univ, Université de Toulon, CNRS, CPT, Marseille, France

⋆ ferrarodario83@gmail.com

## Abstract

We investigate a Fabry-Pérot interferometer in the integer Hall regime in which only one edge channel is transmitted and $n$ channels are trapped into the interferometer loop. Addressing recent experimental observations, we assume that Coulomb blockade effects are completely suppressed due to screening, while keeping track of a residual strong short range electron-electron interaction between the co-propagating edge channels trapped into the interferometer loop. This kind of interaction can be completely described in the framework of the edge-magnetoplasmon scattering matrix theory allowing us to evaluate the backscattering current and the associated differential conductance as a function of the bias voltage. The remarkable features of these quantities are discussed as a function of the number of trapped channels. The developed formalism reveals very general and provides also a simple way to model the experimentally relevant geometry in which some of the trapped channels are absorbed into an Ohmic contact, leading to energy dissipation.

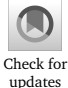

# 1   Introduction

In the last few years various accurate experimental observations shed new light on the remarkable physics associated to the Fabry-Pérot interferometer (FPI) of integer and fractional quantum Hall edge channels [1–3]. Depending on the size of the interference loop two distinct regimes have been achieved. For a small enough loop area, intra-edge interaction plays an essential role and features related to Coulomb blockade occur. In the opposite regime of large central area, the expected Aharonov-Bohm physics of free electrons is recovered in the integer quantum Hall regime. A consistent theoretical interpretation of these results as well as a characterization of the crossover between these two limits has been proposed [4–6]. Various experimental techniques, including gates and ohmic contacts, have been introduced in order to enhance the screening of interaction extending as much as possible the domain of validity of the Aharonov-Bohm regime, where the interaction is negligible and the simple free particles picture seems adequate to properly explain the experimental observations.

However, very recently, new measurements carried out by Choi *et al.* [7,8] have suggested that a richer phenomenology occurs also in this apparently trivial case. In particular, when only one channel is transmitted throughout the FPI, by increasing the number of channels trapped in the loop (namely the integer filling factor of the system), one moves from the standard Aharonov-Bohm effect of electrons (at filling factor $1 \leq \nu \leq 2$) to a more puzzling situation in which a pair of electrons seems to interfere (at filling factor $3 \leq \nu \leq 4$). This phenomenology has been deduced from both the halving of the periodicity of the conductance with respect to the Aharonov-Bohm flux and the doubling of the effective outgoing charge measured through shot noise. These two independent measurements seem to confirm the robustness of the result, moreover various experimental checks have been carried out in order to rule out other possible effects leading to similar phenomenology like the suppression of odd winding of the interferometer with respect to the even ones, leaving the mutual interaction among the channels as the principal responsible of this peculiar pairing effect.

The aim of this paper is to provide the proper theoretical background for the description

of the experimental setup in Ref. [7] in the framework of the edge-magnetoplasmon scattering matrix formalism [9–14] where the two points electron Green's function (first order electron coherence [15, 16]), crucial ingredient to calculate transport properties like conductance and noise, explicitly depends on the transmission of the bosonic mode across the interferometer. As simplest possible case we will assume a strong screened Coulomb interaction and we will investigate carefully the functional form of the scattering matrix as a function of the number of channels trapped into the FPI loop. We will discuss in detail the case of one trapped channel extrapolating then the behavior in case of more channels into the interfering loop. The consequences of the form of these scattering matrices on the current and the conductance will be then investigated. We derive a very powerful and general formalism. On the one hand it allows us to question the simplest academic model of short range strong interaction as a possible way to explain what is observed in experiments, on the other hand it appears suitable to extensions towards more realistic models in which finite length of the interaction and dissipations have been taken into account [17].

The paper is organized as follows. In Section 2 we discuss the edge-magnetoplasmon scattering matrix theory for a FPI as a function of the number of integer Hall edge channels trapped into the loop. We focus in particular on a strong short range ($\delta$-like) screened Coulomb potential. The classical and quantum contributions to the current and the associated differential conductance are derived in Section 3 by means of the Kubo formula. The plots of these quantities, as well as the relevant comments concerning the behavior of the system as a function of the number of trapped channels are reported in Section 4, also in view of a possible interpretation of the experimental observations. In Section 5 we investigate the role played by an ohmic contact absorbing some of the channels, analogous to the one used in realistic setup, in terms of a simple model based only on the energy conservation. Section 6 is devoted to the conclusions, while some technical details of the calculation are discussed in Appendix A.

## 2 Model

### 2.1 Edge-magnetoplasmon description of two interacting channels

Let us start by discussing the physics of two edge channels capacitively coupled along a finite region of length $L$. This problem will be investigated in the framework of the bosonization formalism [18, 19]. Due to the chirality of the two channels we can identify the *incoming region* (1), the *interacting region* (2) and the *outgoing region* (3) (see Fig. 1). We will analyze them in detail in the following.

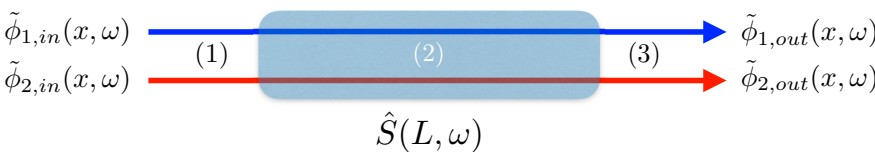

Figure 1: Schematic view of a two integer quantum Hall channels system (filling factor $\nu = 2$). According to the chirality one can easily identify the incoming region (1), the interacting region (2) (shaded area) and the outgoing region (3). In regions (1) and (3) the dynamics of the bosonic fields is well described in terms of free equations of motion, while the outgoing fields are connected to the incoming ones through the edge-magnetoplasmon scattering matrix $\hat{S}(L, \omega)$ which encodes the information of the inter-channel interaction (assumed as strong and short ranged in the main text).

### 2.1.1  Incoming region (1)

In this region the interaction is absent and the Hamiltonian density can be written in term of the Wen's hydrodynamical model [20] ($\hbar = 1$)

$$\mathcal{H}^{(1)} = \frac{v_1}{4\pi}(\partial_x\phi_1)^2 + \frac{v_2}{4\pi}(\partial_x\phi_2)^2. \tag{1}$$

Therefore, one can easily associate a chiral bosonic field to the charge density along each channel according to the conventional prescription [18,19]

$$\rho_i = \frac{1}{2\pi}\partial_x\phi_i \qquad i = 1,2. \tag{2}$$

These bosonic fields propagate freely according to the equations of motion

$$\partial_t\phi_i(x,t) = -v_i\partial_x\phi_i(x,t) \qquad i = 1,2\,, \tag{3}$$

and we have considered different propagation velocities $v_1$ and $v_2$ along the two channels.

### 2.1.2  Interacting region (2)

In this region we assume a density-density short range ($\delta$-like) interaction in such a way that the Hamiltonian density becomes

$$\mathcal{H}^{(2)} = \frac{v_1}{4\pi}(\partial_x\phi_1)^2 + \frac{v_2}{4\pi}(\partial_x\phi_2)^2 + \frac{v_{12}}{2\pi}\partial_x\phi_1\partial_x\phi_2, \tag{4}$$

with $v_{12}$ interaction strength. Notice that, in spite of the fact that high frequency measurements suggest a relevant role played by a finite range of interaction and dissipation, this approximation reveals good at low enough frequencies [17]. According to this, the bosonic fields $\phi_1$ and $\phi_2$ are no longer eigenstates of the Hamiltonian of the system. The equations of motion are then decoupled in terms of a charged and a neutral mode, indicated respectively with $\phi_\rho$ and $\phi_\sigma$. They diagonalize the Hamiltonian with associated eigenvelocities $v_\rho$ and $v_\sigma$ in such a way that the new equations of motion become

$$\partial_t\phi_\eta(x,t) = -v_\eta\partial_x\phi_\eta(x,t) \qquad \eta = \rho,\sigma. \tag{5}$$

Due to the fact that the incoming fields are co-propagating, the above diagonalizing fields are related to $\phi_1$ and $\phi_2$ through a simple rotation of an angle $\theta$ in the field space[1]. This becomes more transparent in the frequency space, namely through a partial Fourier transform with respect to time. Here, one has

$$\begin{aligned}\tilde{\phi}_\rho(x,\omega) &= \cos\theta\,\tilde{\phi}_1(x,\omega) + \sin\theta\,\tilde{\phi}_2(x,\omega)\\ \tilde{\phi}_\sigma(x,\omega) &= -\sin\theta\,\tilde{\phi}_1(x,\omega) + \cos\theta\,\tilde{\phi}_2(x,\omega).\end{aligned} \tag{6}$$

It is worth to note that the angle $\theta$ ($0 \le \theta \le \pi/4$) provides a direct measurement of the strength of the interaction. In particular, $\theta = 0$ corresponds to the non-interacting case, while for $\theta = \pi/4$ one recovers the strong interacting limit.

---

[1]Notice that the velocities $v_\rho$ and $v_\sigma$ are functions of $v_1$, $v_2$ and $v_{12}$ [21]. However, in what follows we are not interested in their explicit functional form, but only in the fact that typically on has $v_\rho \gg v_\sigma$ [10,11].

### 2.1.3   Outgoing region (3)

Analogously to region (1), also in this case inter-channel interaction is negligible and the equations of motion write as in Eq. (3) ($\mathcal{H}^{(1)} = \mathcal{H}^{(3)}$).

The general solution of the above systems of equations can be easily found in the frequency domain and reads

$$\tilde{\phi}_\alpha(x,\omega) = e^{i\frac{\omega}{v_\alpha}(x-x_0)}\tilde{\phi}_\alpha(x_0,\omega),\tag{7}$$

$\tilde{\phi}_\alpha(x_0,\omega)$ being the (possibly frequency dependent) amplitude at the initial condition $x_0$ and where $\alpha = 1,2$ in regions (1) and (3) or $\alpha = \rho, \sigma$ in region (2) respectively.

To completely solve the system we need now to impose the continuity of the fields at the boundaries of the three regions, namely at $x = 0$ and at $x = L$. Notice that, in the Fourier representation we are considering, this is equivalent to impose the conservation of the current across the boundaries.

## 2.2   Open channels

Before investigating the FPI geometry we are interested in, it is useful to recall the expected results in the case of open channels. Here, after some algebra, we obtain the edge-magneto-plasmon scattering matrix representation

$$\begin{pmatrix}\tilde{\phi}_1(L,\omega)\\\tilde{\phi}_2(L,\omega)\end{pmatrix} = \hat{S}(L,\omega)\begin{pmatrix}\tilde{\phi}_1(0,\omega)\\\tilde{\phi}_2(0,\omega)\end{pmatrix},\tag{8}$$

with

$$\hat{S} = \begin{pmatrix}\cos^2\theta e^{i\omega\tau_\rho} + \sin^2\theta e^{i\omega\tau_\sigma} & \sin\theta\cos\theta\left(e^{i\omega\tau_\rho} - e^{i\omega\tau_\sigma}\right)\\\sin\theta\cos\theta\left(e^{i\omega\tau_\rho} - e^{i\omega\tau_\sigma}\right) & \sin^2\theta e^{i\omega\tau_\rho} + \cos^2\theta e^{i\omega\tau_\sigma}\end{pmatrix},\tag{9}$$

and where we have introduced the short-hand notation $\tau_\alpha = L/v_\alpha$ ($\alpha = \rho, \sigma$).

Notice that this result is in full agreement with what is discussed in literature [9,10,12–14] and satisfies the unitarity condition $\hat{S}\cdot\hat{S}^\dagger = \mathbb{I}$ as expected.

## 2.3   One channel trapped in the Fabry-Pérot loop

Thanks to the above results it is now easy to investigate the simplest possible example of the geometry described in Fig. 2, where only one edge channel is trapped into the FPI loop ($n = 1$, $k = 0$). At filling factor $\nu = 2$ (only blue and red channels in Fig. 2) this system represents the natural starting point to model the Choi's experiment of Ref. [7], when only one channel is trapped into the interferometer loop, while the other is transmitted with a tunable amplitude. For sake of generality we will assume two different scattering matrices for the upper ($\hat{S}^{(u)}$) and the lower ($\hat{S}^{(d)}$) part of the interferometer. By properly taking into account the periodic boundary conditions associated to this closed channel one can derive the scattering matrix $\hat{\Sigma}^{(1)}$ for the whole interferometer in the form

$$\begin{pmatrix}\tilde{\phi}_a^{out}\\\tilde{\phi}_b^{out}\end{pmatrix} = \hat{\Sigma}^{(1)}\begin{pmatrix}\tilde{\phi}_a^{in}\\\tilde{\phi}_b^{in}\end{pmatrix},\tag{10}$$

with

$$\hat{\Sigma}^{(1)} = \begin{pmatrix}\dfrac{S_{11}^{(u)} - S_{22}^{(d)} f^{(u)}}{1 - S_{22}^{(u)} S_{22}^{(d)}} & \dfrac{S_{12}^{(u)} S_{12}^{(d)}}{1 - S_{22}^{(u)} S_{22}^{(d)}}\\[2ex]\dfrac{S_{12}^{(u)} S_{12}^{(d)}}{1 - S_{22}^{(u)} S_{22}^{(d)}} & \dfrac{S_{11}^{(d)} - S_{22}^{(u)} f^{(d)}}{1 - S_{22}^{(u)} S_{22}^{(d)}}\end{pmatrix},\tag{11}$$

where we have introduced the phase factor

$$f(\omega) = e^{i\omega(\tau_\rho + \tau_\sigma)}.\tag{12}$$

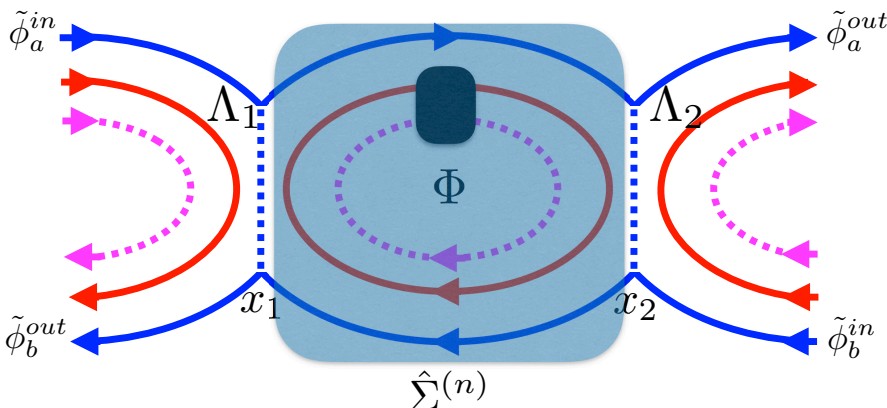

Figure 2: Schematic view of a FPI with $n$ trapped channels in the interfering loop. The action of the interferometer in presence of interaction (assumed strong and short ranged in main text) can be described in terms of the edge-magnetoplasmon scattering matrix $\hat{\Sigma}^{(n)}$. Tunneling of electrons occurs at two quantum point contacts in $x_1$ and $x_2$, respectively with tunneling amplitude $\Lambda_1$ and $\Lambda_2$. The interferometer is also pierced by a flux $\Phi$ of magnetic field which is responsible of the Aharonov-Bohm effect. Moreover, in order to be closer to what is done in experiments, we can consider the presence of an ohmic contact which on the one hand further enhances the screening of the interaction, on the other absorbs the energy of $k \leq n$ trapped channels inducing dephasing.

Notice that the frequency dependence has been omitted for notational convenience.

It is worth to note that the edge-magnetoplasmon scattering matrix $\hat{\Sigma}^{(1)}$ inherits the properties of matrices $\hat{S}^{(u)}$ and $\hat{S}^{(d)}$ and is therefore unitary as expected.

### 2.3.1 Strong interaction limit

In this limit ($\theta = \pi/4$) the expression in Eq. (11) strongly simplifies and one recovers a symmetric configuration $\hat{S}^{(u)} = \hat{S}^{(d)} = \hat{S}$. Moreover, we can safely consider the limit $v_\rho \to +\infty$ ($\tau_\rho \to 0$) for the charged mode.

We then obtain the simple matrix elements

$$S_{11} = S_{22} \approx \frac{1}{2}\left(1 + e^{i\xi}\right), \tag{13}$$

$$S_{12} = S_{21} \approx \frac{1}{2}\left(1 - e^{i\xi}\right), \tag{14}$$

with $\xi = \omega\tau_\sigma$.

Under this approximation, we can then write

$$\hat{\Sigma}^{(1)}(\xi) \approx \begin{pmatrix} 2 - \frac{4}{3}h^{(1)}(\xi) & -1 + \frac{4}{3}h^{(1)}(\xi) \\ -1 + \frac{4}{3}h^{(1)}(\xi) & 2 - \frac{4}{3}h^{(1)}(\xi) \end{pmatrix}, \tag{15}$$

where we have defined

$$h^{(1)}(\xi) = \sum_{n=0}^{+\infty}\left(-\frac{e^{i\xi}}{3}\right)^n = \frac{3}{3 + e^{i\xi}}. \tag{16}$$

Notice that Eq. (16) calls for the possibility of a simple harmonics expansion of the matrix elements of $\hat{\Sigma}^{(1)}$ suitable, as will be clearer in the following, for numerics and useful to easily derive asymptotic behaviors. Moreover, in the zero frequency limit ($\xi = 0$), one directly obtain $\hat{\Sigma}^{(1)}(0) = \mathbb{I}$ as required for a purely capacitive coupling.

## 2.4 Two channels trapped in the Fabry-Pérot loop

In the strong interaction limit discussed above also the case at filling factor $\nu = 3$, where two channels are trapped in the interferometer loop and the last one is transmitted with a tunable amplitude (see Fig. 2), can be easily handled. Without entering into the details of the calculation, also in this case we have a charge mode with a velocity $\nu_\rho$ that is greater with respect to the one of the two neutral modes (assumed $\nu_\sigma$ for both[2]).

Proceeding exactly on the same way as before, after some quite tedious algebra, one obtains the edge-magnetoplasmon scattering matrix

$$\hat{\Sigma}^{(2)}(\xi) \approx \begin{pmatrix} 3 - \frac{12}{5}h^{(2)}(\xi) & -2 + \frac{12}{5}h^{(2)}(\xi) \\ -2 + \frac{12}{5}h^{(2)}(\xi) & 3 - \frac{12}{5}h^{(2)}(\xi) \end{pmatrix}, \tag{17}$$

with

$$h^{(2)}(\xi) = \sum_{n=0}^{+\infty} \left( -\frac{e^{i\xi}}{5} \right)^n = \frac{5}{5 + e^{i\xi}}. \tag{18}$$

Also in this case the zero frequency limit ($\xi = 0$) leads to $\hat{\Sigma}^{(2)}(0) = \mathbb{I}$.

## 3 Current and conductance

We can now discuss the general expression for the current flowing through the system in the framework of the edge-magnetoplasmon scattering matrix formalism and focusing on the weak backscattering regime for both the QPCs in Fig. 2. We will consider the first perturbative order in the backscattering Hamiltonian

$$H_{BS} = \sum_{j=1,2} \Lambda_j \Psi_b^\dagger(x_j)\Psi_a(x_j) + H.c. \,, \tag{19}$$

with $\Psi_a$ and $\Psi_b$ electronic annihilation operators associated to the two edges and $\Lambda_j$ ($j = 1, 2$) tunneling amplitudes associated to the two QPCs. Under this assumption the backscattering current represents a small correction with respect to the quantized Hall current between source and drain. Moreover, this allows us to neglect multiple reflections which could complicates the description of the interferometer. For sake of simplicity the tunneling contributions are assumed to be local, even if more general extended tunneling can be also investigated [23, 24].

The associated backscattering current operator is given by

$$I_B = -e \sum_{j=1,2} i\Lambda_j \Psi_b^\dagger(x_j)\Psi_a(x_j) + H.c. \tag{20}$$

In order to evaluate the average value of the backscattering current we can use, as usual, the Kubo formula [25]

$$\langle I_B \rangle = -i \int_{-\infty}^{t} dt' \langle \left[ I_B(t), H_{BS}(t') \right] \rangle \tag{21}$$

where operators are written in the interaction picture, namely evolved in time according to the free edge Hamiltonian only. Notice that all the averages discussed above are taken with respect to the ground state of the free bosonic systems in absence of backscattering.

In order to evaluate the backscattering current in Eq. (21) as a function of the elements of the edge-magnetoplasmon scattering matrices derived in the previous section we need to

---

[2]Notice that this symmetry is reminiscent of the analogous hidden symmetry observed for the states belonging to the Jain's sequence of the fractional quantum Hall effect [22].

recall the fact that the fermionic annihilation operator can be seen as a coherent state of edge-magnetoplasmon in the form

$$\Psi(x) = \frac{1}{\sqrt{2\pi\alpha}} e^{i\phi(x)}, \tag{22}$$

$\alpha$ a finite-length cut-off. [18]

Because the calculation of the averaged backscattering current naturally involves four-vertex operators it is useful to introduce the general correlator

$$\langle e^{-iA} e^{iB} e^{-iC} e^{iD} \rangle = \exp\left\{ -\frac{1}{2} \left[ \langle A^2 \rangle + \langle B^2 \rangle + \langle C^2 \rangle + \langle D^2 \rangle \right] \right.$$
$$\left. + \langle AB \rangle - \langle AC \rangle + \langle AD \rangle + \langle BC \rangle - \langle BD \rangle + \langle CD \rangle \right\}, \tag{23}$$

based on the Baker-Campbel-Hausdorff formula and the Wick theorem applied to the bosonic fields and valid for arbitrary gaussian fields $A$, $B$, $C$ and $D$ those commutation relations lead to complex functions.

By properly taking into account the action of the edge-magnetoplasmon scattering matrix and assuming the same propagation velocity for both free channels ($v_1 = v_2 = v$) we can evaluate explicitly all the contributions to the current. Notice that the effect of a bias difference between the edge $b$ (at voltage $V$) and the edge $a$ (grounded) is taken into account through the standard Peierls substitution [26] $\Psi_b(x, t) \to e^{-i\omega_0 t} \Psi_b(x, t)$ with $\omega_0 = eV/\hbar$ and where the field $\Psi_a(x, t)$ is left untouched.

The averaged current can be naturally separated into a classical contribution, diagonal in the QPCs action, and a quantum contribution, which is the off diagonal interference term. In the following we will discuss these terms in detail mainly focusing on the zero temperature limit.

## 3.1 Classical contributions to the current

For what it concerns the classical contributions to the current, namely the one diagonal in the QPCs tunneling amplitudes, one obtains

$$I_B^{cl} = \frac{e}{2\pi^2 \alpha^2} \left[ |\Lambda_1|^2 + |\Lambda_2|^2 \right] \int_{-\infty}^{+\infty} dz \sin(\omega_0 z) \Im\left[ \mathscr{G}^{(l)}(z) \right], \tag{24}$$

with

$$\mathscr{G}^{(l)}(z) = \exp\left\{ -2 \int_0^{+\infty} \frac{d\omega}{\omega} \left\{ 1 - \Re\left[ \Sigma_{ab}^{(l)}(\omega) \right] \right\} \left[ 1 - e^{-i\omega z} \right] \right\}, \tag{25}$$

and where $\Re[...]$ and $\Im[...]$ indicate respectively the real and the imaginary part. Notice that the above formula can be applied to both the case of one ($l = 1$) and two ($l = 2$) trapped channels in the FPI. The conventional non interacting case is easily recovered by neglecting the term associated to the edge-magnetoplasmon scattering matrix ($\Sigma_{ab}^{(0)} = 0$). It is worth to point out the fact that, due to translational invariance in the time domain, the expression in Eq. (24) is indeed independent of time.

As expected, the above term does not depend on the position of the QPCs and on the flux of magnetic field piercing the interferometer. It corresponds to the sum of the contributions associated to two distinct single QPC geometries, proportional respectively to $|\Lambda_1|^2$ and $|\Lambda_2|^2$.

By replacing the explicit form of the scattering matrix elements respectively for $\hat{\Sigma}^{(1)}$ and $\hat{\Sigma}^{(2)}$ (see Eq. (15) and Eq. (17)) one obtains the factorization

$$\mathscr{G}^{(l)}(z) = \mathscr{G}_{int}^{(l)}(z) \mathscr{G}_{free}(z) \qquad l = 1, 2, \tag{26}$$

with (see Appendix A)

$$\mathscr{I}_{free}(z) = \exp\left\{ -2 \int_0^\infty \frac{d\omega}{\omega} \left[ 1 - e^{-i\omega z} \right] e^{-\omega/\omega_\rho} \right\} \tag{27}$$

$$= \frac{1}{(1 + i\omega_\rho z)^2}, \tag{28}$$

the free fermion contribution and

$$\mathscr{I}_{int}^{(1)}(z) = \exp\left\{ -2 \int_0^\infty \frac{d\omega}{\omega} \left( \frac{\cos\omega\tau_\sigma - 1}{5 + 3\cos\omega\tau_\sigma} \right) \left[ 1 - e^{-i\omega z} \right] e^{-\omega/\omega_\rho} \right\}, \tag{29}$$

$$\mathscr{I}_{int}^{(2)}(z) = \exp\left\{ -8 \int_0^\infty \frac{d\omega}{\omega} \left( \frac{\cos\omega\tau_\sigma - 1}{13 + 5\cos\omega\tau_\sigma} \right) \left[ 1 - e^{-i\omega z} \right] e^{-\omega/\omega_\rho} \right\}, \tag{30}$$

the corrections due to the interaction. Notice that, for further convenience, we have introduced the convergence factor $e^{-\omega/\omega_\rho}$ with high frequency cut-off $\omega_\rho = v_\rho/\alpha$ in order to have well behaved integrals for $\omega \to +\infty$.

## 3.2 Quantum contribution to the current

Differently from the classical contribution discussed above, this term depends non locally on the two QPCs amplitudes and encodes information about the Aharonov-Bohm interference at the level of the FPI. It is given by

$$I_B^q = \frac{e}{\pi^2 \alpha^2} \int_{-\infty}^{+\infty} dz \, \Im\left[ \Lambda_1 \Lambda_2^* e^{i\omega_0 z} \mathscr{W}^{(l)} \right] \Im\left[ \mathscr{Y}^{(l)}(\Delta t - z) \right], \tag{31}$$

with $\Delta t = (x_1 - x_2)/v$,

$$\mathscr{W}^{(l)} = \exp\left\{ \int_0^{+\infty} \frac{d\omega}{\omega} \left[ \Sigma_{ab}^{(l)}(\omega) + \Sigma_{ba}^{(l)}(\omega) \right] \right\} \tag{32}$$

and

$$\mathscr{Y}^{(l)}(z) = \exp\left\{ -\int_0^{+\infty} \frac{d\omega}{\omega} \left[ 2 - \left( \Sigma_{aa}^{*(l)}(\omega) + \Sigma_{bb}^{(l)}(\omega) \right) e^{i\omega z} \right] \right\}. \tag{33}$$

In analogy with what discussed for the classical current, also this contribution does not depend explicitly on time.

In terms of the explicit form of the scattering matrix elements for $\hat{\Sigma}^{(1)}$ and $\hat{\Sigma}^{(2)}$ (Eq. (15) and Eq. (17)), one has

$$\mathscr{W}^{(1)} = \exp\left\{ 2 \int_0^\infty \frac{d\omega}{\omega} \left( \frac{1 - e^{i\omega\tau_\sigma}}{3 + e^{i\omega\tau_\sigma}} \right) e^{-\omega/\omega_\rho} \right\}, \tag{34}$$

$$\mathscr{W}^{(2)} = \exp\left\{ 4 \int_0^\infty \frac{d\omega}{\omega} \left( \frac{1 - e^{i\omega\tau_\sigma}}{5 + e^{i\omega\tau_\sigma}} \right) e^{-\omega/\omega_\rho} \right\}. \tag{35}$$

Moreover, also in this case we can factorize the interaction contribution in the form $\mathscr{Y}^{(l)}(z) = \mathscr{Y}_{int}^{(l)}(z) \mathscr{I}_{free}(-z)$ with,

$$\mathscr{Y}_{int}^{(1)}(z) = \exp\left\{ 2 \int_0^\infty \frac{d\omega}{\omega} \left( \frac{\cos\omega\tau_\sigma - 1}{5 + 3\cos\omega\tau_\sigma} \right) e^{i\omega z} e^{-\omega/\omega_\rho} \right\}, \tag{36}$$

$$\mathscr{Y}_{int}^{(2)}(z) = \exp\left\{ 8 \int_0^\infty \frac{d\omega}{\omega} \left( \frac{\cos\omega\tau_\sigma - 1}{13 + 5\cos\omega\tau_\sigma} \right) e^{i\omega z} e^{-\omega/\omega_\rho} \right\}. \tag{37}$$

According to the Fourier series expansion of the functions $h^{(1)}$ (Eq. (16)) and $h^{(2)}$ (Eq. (18)) it is possible to derive useful asymptotic limits for the functions $\mathscr{J}_{int}^{(l)}$, $\mathscr{W}^{(l)}$ and $\mathscr{Y}_{int}^{(l)}$ ($l = 1, 2$) which are discussed in detail in Appendix A.

In particular, focusing on the role played by $\mathscr{W}^{(l)}$, Eq. (31) can be approximated in a very good way by ($j = 0, 1, 2$)

$$I_B^q(\Phi, \omega_0) \approx -\frac{e|\Lambda_1||\Lambda_2|}{\pi^2 \alpha^2}(\omega_\rho \tau_\sigma)^{a_j} e^{-\gamma_j} \cos(\omega_0 \Delta t - \Phi - \theta_j) \times$$
$$\int_{-\infty}^{+\infty} dz \sin \omega_0 z \, \Im\left[\mathscr{Y}_{int}^{(j)}(z) \mathscr{J}_{free}(-z)\right] \quad (38)$$

where we have included the conventional free fermion case $j = 0$. In the above expression we have introduced the compact notation $a_0 = 0$, $a_1 = 2/3$ and $a_2 = 4/5$; $\theta_0 = 0$, $\theta_1 = \pi/3$ and $\theta_2 = 2\pi/5$; $\gamma_0 = 0$, $\gamma_1 \approx 0.129656$ and $\gamma_2 \approx 0.099524$ (see Appendix A). Moreover, it is worth to note that in the non interacting case we trivially have $\mathscr{Y}_{int}^{(0)} = 1$. We observe that the more dramatic effects associated to the presence of the trapped channels consists in modifying the scaling of the tunneling amplitude through a cut-off dependent contribution and to add an additional phase shift. Both these contributions crucially depend on the number of trapped channels.

## 4 Results

### 4.1 Classical contribution

In Fig. 3 we show the behavior of the classical contribution to the current and the associated differential conductance

$$G_B^{cl} \propto \frac{\partial \langle I_B^{cl} \rangle}{\partial \omega_0} \quad (39)$$
$$= \frac{e}{2\pi^2 \alpha^2}\left[|\Lambda_1|^2 + |\Lambda_2|^2\right]\int_{-\infty}^{+\infty} dz z \cos(\omega_0 z) \Im\left[\mathscr{J}^{(l)}(z)\right]. \quad (40)$$

In order to take into account the renormalization of the tunneling amplitudes induced by the presence of the trapped channels and to better compare the results, the curves are normalized here with respect to the proper cut-off dependent factor $(\omega_\rho \tau_\sigma)^{a_j} e^{-\gamma_j}$ ($j = 0, 1, 2$).

Concerning the current (top panel of Fig. 3) all curves are linear as a function of the Josephson frequency (and consequently of the voltage $V$) for $\omega_0 \approx 0$ as a consequence of the fact that the edge-magnetoplasmon scattering matrix reduces to the identity at low enough frequency. However, the interacting cases (dashed blue and dotted-dashed red curves) strongly deviate from this linearity at higher voltages ($\omega_0 \tau_\sigma \approx \pi$) and show a remarkable oscillating behavior. This is even more evident for what it concerns the differential conductance (bottom panel of Fig. 3) which is constant (up to a small deviations due to the high frequency cut-off used in the numerics) in the free case, but oscillates and decay quite fast by increasing the number of trapped channels. Notice that this phenomenology is reminiscent of what derived in literature for a FPI in the integer and fractional quantum Hall regime [27] or for the topological insulators in presence of interaction [28].

**SciPost** SciPost Phys. 3, 014 (2017)

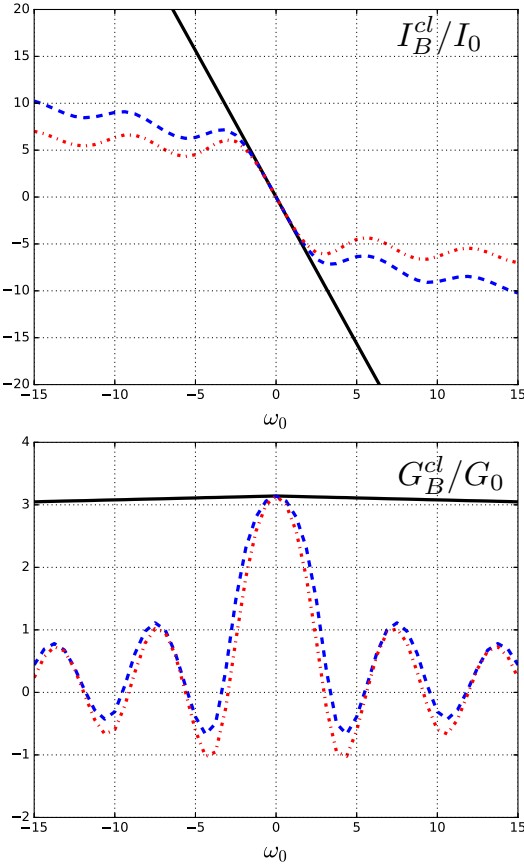

Figure 3: Classical current in units of $I_0 = e|\Lambda|^2/(\pi \nu_\rho \tau_\sigma)^2$ (top) and differential conductance in units of $G_0 = eI_0$ (bottom). Every curve is properly further rescaled with respect to the factor $(\omega_\rho \tau_\sigma)^{a_j} e^{-\gamma_j}$ in order to keep track of the interaction induced renormalization of the tunneling amplitudes for the non interacting case ($j = 0$, full black curve), the one trapped channel case ($j = 1$, dashed blue curve) and the two trapped channels case ($j = 2$, dotted-dashed red curve) as a function of the Josephson frequency $\omega_0 = eV_0/\hbar$. Parameters are: $\tau_\sigma = 1$, $\omega_\rho \tau_\sigma = 1000$, $|\Lambda_1| = |\Lambda_2| = |\Lambda|$ and $\hbar = 1$.

## 4.2 Quantum contribution to the current

Concerning the interference contribution, relevant information about the physics of the systems can be extracted form the conductance calculated at zero voltage ($j = 0, 1, 2$.)

$$G_B^q(\Phi, \omega_0 = 0) \propto \frac{\partial I_B^q}{\partial \omega_0}\big|_{\omega_0 = 0} \tag{41}$$

$$\approx -\frac{e|\Lambda_1||\Lambda_2|}{\pi^2 \alpha^2}(\omega_\rho \tau_\sigma)^{a_j} e^{-\gamma_j} \cos\left(\Phi + \theta_j\right) \int_{-\infty}^{+\infty} dz z \Im\left[\mathscr{Y}_{int}^{(j)}(z)\mathscr{J}_{free}(-z)\right] \tag{42}$$

The behavior of this quantity as a function of the Aharonov-Bohm phase $\Phi$ is represented in Fig. 4, where the phase shifts $\theta_1 = \pi/3$ and $\theta_2 = 2\pi/5$ induced by the presence of a different number of channels trapped into the interferometer loop are clearly visible.

It is worth to note that the various curves have the same amplitude. This can be easily found by comparing the non interacting case ($j = 0$), where the integral in Eq. (42) can be

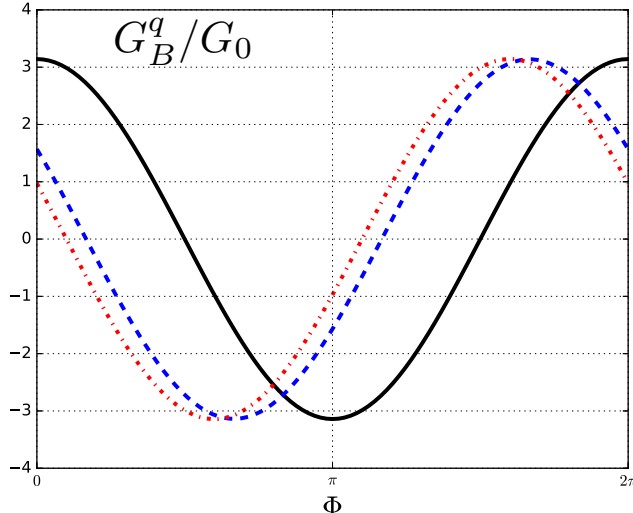

Figure 4: Quantum contribution to the differential conductance in units of $G_0$, calculated at $\omega_0 = 0$ as a function of the Aharonov-Bohm phase ($\Phi$). Also in this case every curve is properly further rescaled with respect to the factor $(\omega_\rho \tau_\sigma)^{a_j} e^{-\gamma_j}$ in order to keep track of the interaction induced renormalization of the tunneling amplitudes for the non interacting case ($j = 0$, full black curve), the one trapped channel case ($j = 1$, dashed blue curve) and the two trapped channels case ($j = 2$, dotted-dashed red curve). Parameters are: $\tau_\sigma = 1$, $\omega_\rho \tau_\sigma = 1000$, $|\Lambda_1| = |\Lambda_2| = |\Lambda|$ and $\hbar = 1$.

evaluated analytically according to the relation

$$\int_{-\infty}^{+\infty} dz z \Im \left[ \frac{1}{(1 + i\omega_\rho z)^2} \right] = -\frac{\pi}{\omega_\rho^2}, \tag{43}$$

due to the second order poles in the complex plane and coincide with what obtained numerically for the interacting cases ($j = 1, 2$). This is a direct consequence of the zero energy properties of the edge-magnetoplasmon scattering matrix which guarantee the fact that the electrons are effectively free at low enough frequency.

## 4.3 Considerations about multiple electron tunneling

As discussed above, Choi's experiment shows a puzzling and extremely interesting evolution of the effective tunneling charge as a function of the number of trapped channels. In order to shed light on this behavior one can imagine to allow also the tunneling of excitations with $m$ times the charge of an electron[3] ($m \in \mathbb{N}$) at the QPCs, despite the fact that these multiple excitations are less relevant with respect to the electrons in the renormalisation group sense [36]. The vertex operator associated to them is given by

$$\Psi^{(m)}(x) \propto e^{im\varphi(x)}, \tag{44}$$

with the corresponding tunneling Hamiltonian

$$H_T^{(m)} = \sum_{j=1,2} \Lambda_j^{(m)} e^{im\varphi_b(x_j)} e^{-im\varphi_a(x_j)} + H.c. \tag{45}$$

---

[3]Notice that a similar analysis involving the tunneling of multiple fractionally charged excitation has been proposed in Refs. [29–32] as a possible explanation for the unexpected evolution of the effective charge in a QPC geometry in the composite edge states of the fractional quantum Hall effect [33–35].

In this case, as far as we consider a gaussian model for the edge states dynamics, the expressions for this kind of excitations can be derived directly from the one for the electrons through the substitutions $\mathscr{X} \to [\mathscr{X}]^{m^2}$, being $\mathscr{X} = \mathscr{J}^{(1,2)}, \mathscr{W}^{(1,2)}$ or $\mathscr{Y}^{(1,2)}$ and with a consequent replacement of the tunneling amplitudes and the charge associated to the excitations. This leads to a superohmic behavior which reflects on the fact that the conductance at zero bias is identically zero. This fact can be verified analytically in the non interacting case due to the relation

$$\int_{-\infty}^{+\infty} dz z \Im\left[\frac{1}{(1 + i\omega_\rho z)^{2m^2}}\right] = 0 \quad \text{for} \quad m \neq 1, \tag{46}$$

which is a consequence of the presence of higher order poles in the complex plane and numerically in the other cases.

Therefore, no signature associated to higher charge carrier are expected in the framework of the proposed model. According to this, the comparison with experimental observations seems to question the simple picture based on short range strong interaction usually considered as a valuable work hypothesis and suggests that a more involved description is required. In particular dissipative effects at the level of the edge-magnetoplasmon scattering matrix [17], finite range interaction and non-gaussianity in the particle injection process [10,37] could play a relevant role.

# 5 Effects of an ohmic contact in the interferometer loop

In order to be further closer to what has been done in experiments, we can also consider a modified set-up in which an integer number $k \leq n$ of the trapped channels is absorbed by an ohmic contact. This configuration has been introduced in order to both enhancing the screening of the interaction and inducing dephasing among the trapped channels (see Fig. 1) [7]. We will study this geometry in the full equilibration limit ($\tau_\sigma \to +\infty$) where we can neglect the oscillations appearing in the edge-magnetoplasmon scattering matrices (see Eqs. (15) and (17)). This leads to a semiclassical approximation based only on the conservation of a unitary energy flow. In full generality it writes

$$\hat{\Sigma}_\uparrow^{(n,k)} = \begin{pmatrix} \frac{(n-k+1)(n+1)}{n^2(k+2)+3n+1} & \frac{(n-k+1)n}{n^2(k+2)+3n+1} \\ \frac{(n-k+1)n}{n^2(k+2)+3n+1} & \frac{(n+1)^2}{n^2(k+2)+3n+1} \end{pmatrix}. \tag{47}$$

The notation $\uparrow$ indicates an ohmic contact placed in the upper part of the interferometer loop (see Fig. 2), as opposite to $\downarrow$ for an ohmic contact in the lower part whose scattering matrix elements are obtained through the replacements $\uparrow \leftrightarrow \downarrow$, $a \leftrightarrow b$. Notice that the position of the ohmic contact does not affect the results concerning the current flowing across the sample.

Depending on the incoming arm of injection ($a$ or $b$) one can have two different fraction of the unitary energy flux leaking into the ohmic contact, namely

$$\Delta E_{a,\uparrow} = 1 - \Sigma_{aa,\uparrow}^{(n,k)} - \Sigma_{ba,\uparrow}^{(n,k)} = \frac{k(n+1)^2}{n^2(k+2)+3n+1},$$

$$\Delta E_{b,\uparrow} = 1 - \Sigma_{ab,\uparrow}^{(n,k)} - \Sigma_{bb,\uparrow}^{(n,k)} = \frac{kn(n+1)}{n^2(k+2)+3n+1}.$$

In absence of ohmic contact ($k = 0$) the scattering matrix in Eq. (47) reduces to

$$\hat{\Sigma}_\uparrow^{(n,0)} = \hat{\Sigma}_\downarrow^{(n,0)} = \hat{\Sigma}^{(n,0)} = \begin{pmatrix} \frac{n+1}{2n+1} & \frac{n}{2n+1} \\ \frac{n}{2n+1} & \frac{n+1}{2n+1} \end{pmatrix}, \tag{48}$$

with an obvious zero energy leakage ($\Delta E_{a,\uparrow} = \Delta E_{b,\uparrow} = 0$). Two important comments are in order at this point. First of all, we notice that it is possible to obtain the expressions for $\hat{\Sigma}^{(1,0)}$ and $\hat{\Sigma}^{(2,0)}$ directly from Eq. (15) and (17) in the semiclassical limit, where all the oscillating terms are neglected. Eq. (48) represents therefore a generalization of what is done above as far as equilibration among the channels comes into play [38]. Moreover, this fact also represents a validation of the renormalisation of the tunneling amplitudes previously derived, due to the fact that the dominant contribution to the integrals comes indeed from a region where the considered approximation holds.

According to the previous considerations, the classical and quantum contribution to the current can be written, in terms of the elements of the scattering matrix and the energy loss, as

$$I_B^{cl} = \frac{e}{2\pi^2 \alpha^2} \left( |\Lambda_1|^2 + |\Lambda_2|^2 \right) \mathscr{R} \left[ \omega_0, \omega_\rho, \Sigma_{aa}^{(n,k)} + \Sigma_{bb}^{(n,k)} + \Delta E_a + \Delta E_a \right] \tag{49}$$

$$I_B^q = -\frac{e}{\pi^2 \alpha^2} \left( |\Lambda_1||\Lambda_2| \right) \cos\left( \omega_0 \Delta t - \Phi \right) \exp\left\{ -(\Delta E_a + \Delta E_b) \int_0^{+\infty} \frac{d\omega}{\omega} e^{-\omega/\omega_\rho} \right\} \times$$
$$\mathscr{R} \left[ \omega_0, \omega_\rho, \Sigma_{aa}^{(n,k)} + \Sigma_{bb}^{(n,k)} \right] \tag{50}$$

with

$$\mathscr{R}\left[ x, y, \alpha \right] = \frac{2\pi}{\Gamma(\alpha)} \frac{|x|^{\alpha-1}}{y^\alpha} e^{-|x|/y} \mathrm{sgn}(x), \tag{51}$$

and $\Gamma(\alpha)$ the Euler's Gamma function. Notice that we have omitted the indication about the placement of the ohmic contact due to the fact that the results are independent of this because of the symmetries relating $\hat{\Sigma}_\uparrow^{(n,k)}$ and $\hat{\Sigma}_\downarrow^{(n,k)}$.

In absence of ohmic contact ($\Delta E_a = \Delta E_b = 0$), but still considering the equilibrated limit, one obtains (for $|\Lambda_1| = |\Lambda_2| = |\Lambda|$)

$$I_B^{cl} + I_B^q = \frac{e|\Lambda|^2}{\pi^2 \alpha^2} \left[ 1 - \cos\left( \omega_0 \Delta t - \Phi \right) \right] \mathscr{R} \left[ \omega_0, \omega_\rho, \Sigma_{aa}^{(n,0)} + \Sigma_{bb}^{(n,0)} \right], \tag{52}$$

showing oscillation as a function of the piercing magnetic flux modulating the overall power-law behavior $\omega_0^{\Sigma_{aa}^{(n,0)} + \Sigma_{bb}^{(n,0)} - 1}$ reminiscent of the one observed for the fractional quantum Hall effect [27] and topological insulators in presence of interaction [28]. It is worth to note that the exponents satisfies

$$1 < \Sigma_{aa}^{(n,0)} + \Sigma_{bb}^{(n,0)} < 2, \tag{53}$$

leading to a sub-ohmic and not diverging current-voltage characteristic.

The presence of an ohmic contact dramatically affects the quantum contribution, which is exponentially suppressed according to the vanishing prefactor in Eq. (50). Indeed, the integral at the exponent is formally divergent and have to be regularized by adding an additional low frequency cut-off depending on the typical equilibration length of the interferometer loop [39]. However, this exponential suppression still survives at finite temperature even after having taken care of this formal divergence.

Moreover, the classical and quantum contribution to the current scale in a different way as a function of the voltage, namely

$$I_B^{cl} \propto \omega_0^{\Sigma_{aa}^{(n,k)} + \Sigma_{bb}^{(n,k)} + \Delta E_a + \Delta E_a - 1}, \tag{54}$$

$$I_B^q \propto \omega_0^{\Sigma_{aa}^{(n,k)} + \Sigma_{bb}^{(n,k)} - 1}. \tag{55}$$

## 6  Conclusion

In the present paper we have discussed the physics of a FPI in the integer quantum Hall regime. Motivated by very recent experiments, we focused on a system at filling factor $\nu = n + 1$ ($n \in \mathbb{N}$) where only one edge channel is transmitted across the sample, while the other $n$ are trapped into the interferometer loop. Due to screening, the only residual interaction effect is given by a strong short range inter-channel interaction that we described in terms of the edge-magnetoplasmon scattering matrix formalism. The major effects associated to the presence of interacting trapped channels are: a renormalisation of the tunneling amplitudes affecting in exactly the same way the classical and the quantum contribution to the current, an evident damped and oscillatory behavior of the differential conductance in contrast to the constant (ohmic) behavoir observed in absence of trapped channels and an additional phase shift in the Aharonov-Bohm periodicity that is clearly visible in the differential conductance at zero bias. We have also discussed a simple model for a system in which $k$ ($0 \leq k \leq n$) channels in the loop are absorbed by an ohmic contact, based only on the energy conservation. Here we observed that, as long as the energy is not conserved due to losses into the contact, the quantum contribution to the conductance is strongly suppressed and only the classical contribution survives, showing a non-universal power-law behavior reminiscent of the element of the effective edge-magnetoplasmon scattering matrix. It is worth to note that the comparison between our analysis and the experiments allows to doubt about this simple and widely accepted model for the inter-edge channel interaction as the proper description of the considered set-up. According to this, additional physical effects like dissipation or non-gaussianity could play a major role. However, the developed formalism is general enough to allow extention towards these directions.

## Acknowledgements

We thanks I. P. Levkivskyi, E. Idrisov, A. Borin and A. Goremykina for useful discussions. We acknowledge the financial support of Swiss National Science Foundation. D. F. want to acknowledge the support of Grant No. ANR-2010-BLANC-0412 ("1 shot") and of ANR-2014-BLANC "one shot reloaded" is acknowledged. Part of this work was carried out in the framework of Labex ARCHIMEDE Grant No. ANR-11-LABX-0033 and of A*MIDEX project Grant No. ANR-11-IDEX-0001-02, funded by the "investissements d'avenir" French Government program managed by the French National Research Agency (ANR).

## A  Consideration about the integrals

Aim of this Appendix is to investigate the asymptotic behavior of the functions $\mathscr{I}_{int}^{(l)}$, $\mathscr{W}^{(l)}$ and $\mathscr{Y}_{int}^{(l)}$ ($l = 1, 2$) defined in the main text. According to the integral representation

$$f(\eta, A) = \int_0^{+\infty} \frac{d\omega}{\omega} \left(1 - e^{-i\omega\eta}\right) e^{-\omega A} = \ln\left(1 + i\frac{\eta}{A}\right), \tag{56}$$

and recalling the Fourier series expansion of $h^{(1)}$ and $h^{(2)}$ in Eqs. (16) and (18) one obtains

$$\mathcal{J}_{int}^{(1)}(z) = \exp\left\{\frac{4}{3}\sum_{n=0}^{+\infty}\frac{(-1)^n}{3^n}\left[f\left(z,\frac{1}{\omega_\rho}-ni\tau_\sigma\right)+f\left(z,\frac{1}{\omega_\rho}+ni\tau_\sigma\right)\right]-2f\left(z,\frac{1}{\omega_\rho}\right)\right\},$$
(57)

$$\mathcal{J}_{int}^{(2)}(z) = \exp\left\{\frac{12}{5}\sum_{n=0}^{+\infty}\frac{(-1)^n}{5^n}\left[f\left(z,\frac{1}{\omega_\rho}-ni\tau_\sigma\right)+f\left(z,\frac{1}{\omega_\rho}+ni\tau_\sigma\right)\right]-4f\left(z,\frac{1}{\omega_\rho}\right)\right\},$$
(58)

$$\mathcal{W}^{(1)} = \exp\left\{\frac{2}{3}\sum_{n=0}^{\infty}\frac{(-1)^n}{3^n}f\left(-\tau_\sigma,\frac{1}{\omega_\rho}-ni\tau_\sigma\right)\right\},$$
(59)

$$\mathcal{W}^{(2)} = \exp\left\{\frac{4}{5}\sum_{n=0}^{\infty}\frac{(-1)^n}{5^n}f\left(-\tau_\sigma,\frac{1}{\omega_\rho}-ni\tau_\sigma\right)\right\},$$
(60)

$$\mathcal{Y}_{int}^{(1)}(z) = \exp\left\{\frac{4}{3}\sum_{n=0}^{+\infty}\frac{(-1)^n}{3^n}\left[f\left(-n\tau_\sigma,\frac{1}{\omega_\rho}-iz\right)+f\left(n\tau_\sigma,\frac{1}{\omega_\rho}-iz\right)\right]\right\},$$
(61)

$$\mathcal{Y}_{int}^{(2)}(z) = \exp\left\{\frac{12}{5}\sum_{n=0}^{+\infty}\frac{(-1)^n}{5^n}\left[f\left(-n\tau_\sigma,\frac{1}{\omega_\rho}-iz\right)+f\left(n\tau_\sigma,\frac{1}{\omega_\rho}-iz\right)\right]\right\}.$$
(62)

The above series converge quite fast and are helpful both in the numeric evaluation of the integrals for the current and in order to obtain asymptotic expressions under the natural condition $\omega_\rho\tau_\sigma \gg 1$.

## A.1 $\mathcal{J}_{int}^{(1)}$ and $\mathcal{J}_{int}^{(2)}$

In the limit $|z| \gg n\tau_\sigma$ ($n \in \mathbb{N}$) one directly obtains

$$\mathcal{J}_{int}^{(1)}(|z| \to +\infty) \approx (\omega_\rho\tau_\sigma)^{\frac{2}{3}}e^{-\gamma_1},$$
(63)

$$\mathcal{J}_{int}^{(2)}(|z| \to +\infty) \approx (\omega_\rho\tau_\sigma)^{\frac{4}{5}}e^{-\gamma_2},$$
(64)

with

$$\gamma_1 = \frac{8}{3}\sum_{n=1}\left(-\frac{1}{3}\right)^n\log n \approx 0.129656,$$
(65)

$$\gamma_2 = \frac{24}{5}\sum_{n=1}\left(-\frac{1}{5}\right)^n\log n \approx 0.099524.$$
(66)

## A.2 $\mathcal{W}^{(1)}$ and $\mathcal{W}^{(2)}$

In this case there is no dependence on $z$, but it is still possible to resum the series in an approximate way obtaining

$$\mathcal{W}^{(1)} = (\omega_\rho\tau_\sigma)^{\frac{2}{3}}e^{-i\frac{\pi}{3}}e^{-\gamma_1},$$
(67)

$$\mathcal{W}^{(2)} = (\omega_\rho\tau_\sigma)^{\frac{4}{5}}e^{-i\frac{2\pi}{5}}e^{-\gamma_2}.$$
(68)

Notice that the appearance of a phase factor that will enter directly as a shift of the magnetic flux into the quantum contribution to the current.

## A.3 $\mathscr{Y}_{int}^{(1)}$ and $\mathscr{Y}_{int}^{(2)}$

Here, in the limit $|z| \gg n\tau_\sigma$ ($n \in \mathbb{N}$) these functions reduce to

$$\mathscr{Y}_{int}^{(1)}(|z| \to +\infty) = \mathscr{Y}_{int}^{(2)}(|z| \to +\infty) \approx 1. \tag{69}$$

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
