# Peer review of "Interaction effects in a multi-channel Fabry-Pérot interferometer in the Aharonov-Bohm regime"

_SciPost Physics, doi:SciPost Phys. 3, 014 (2017)_

## Round 2 · Referee Report · Anonymous · 2017-6-1

Strengths

1. The authors attempt to explain recent experimental observations by the group of M. Heiblum of "electron pairing" in the IQHE regime.
2. The paper presents a microscopic calculation based on a short-range interacting bosonic model using scattering approach.

Weaknesses

1. The model does not seem to account for the observation of Ref. [6].
2. It is unclear what one can learn from this paper either from physics, or from methodology perspective.
3. There is no clear discussion of the results, and there is no comparison with experiments.
4. The semiclassical calculation in the presence of the Ohmic contact is not clearly explained and/or justified.
5. There is no discussion of what one could expect going beyond short-range interacting model.
6. The integral in Eq. 49 is divergent. Is there a typo?
7. The quality of English is rather poor and needs to be improved.

Report

This paper attempts to explain the observations of Ref. [6] of "electron pairing" in electronic Fabry-Perot interferometers. It is an interesting theoretical problem, and the physical mechanism for this pairing is unclear at the moment. The authors build on their previous work on electronic interferometers in the IQHE regime, by using a short-range interacting model of edge-states to describe the experimental setup of Ref. [6]. However, this model does not seem to account for experimental observations.

While the paper is worth publishing in some form after substantial changes, I do not feel that it clears the bar for SciPost in terms of its scientific quality.

Requested changes

1. The list of changes is related to Weaknesses, see above.

---

## Round 2 · Referee Report · Anonymous · 2017-7-16

Strengths

the study is motivated by a recent unexplained experiment from the Heiblum group
the authors study a model which seems suitable to describe the experiment
the authors use an innovative formalism to solve their model

Weaknesses

the manuscript does not explain the experiment the authors want to describe
the conclusion about the model being unable to describe the experiment may be too strong

Report

Referee report on the manuscript “Interaction effects in a multi-channel Fabry-Pérot interferometer in the Aharonov-Bohm regime” by D. Ferraro, E. Sukhorukov

In their manuscript the authors study interaction effects in a Fabry-Perot quantum Hall interferometer operating in the integer regime, with a screened interaction between bulk and edges. The study is motivated by a recent experiment from the Heiblum group, which found an unexpected magnetic field periodicity of the interference signal, which together with shot noise measurements was taken to indicate electron pairing in such an interferometer. Like in the experiment, the present authors study backscattering of the outermost edge channel, with an additional n channels trapped inside the interferometer. Due to the presence of a screening contact, the authors assume the interaction between interfering edge and inner edge channels to be short-ranged, which should be realistic. It seems to me that the authors may consider the right type of model to describe the experiment.

The authors study their model using bosonization, and combine a non-perturbative scattering formalism for the boson fields with the usual expression for Fermi operators as exponentials of the bosons. I find this approach innovative and interesting, and believe that it is a powerful tool to study fluctuation effects in the Fabry-Perot geometry. It seems to me that the authors display considerable skill in evaluating the technically demanding integrals, and manage to discuss their results in a clear way.

Somewhat disappointingly the calculation presented in the manuscript does not manage to explain the experimental findings. The authors interpret this state of affairs as indicating that the model they study can be ruled out for an explanation of the experimental findings. I believe this conclusion is too strong and needs to be softened in order to make the manuscript publishable in SciPost. First, there is always the possibility that something is missed in an analysis. Second, on a recent conference I have heard the claim that the model studied in the present manuscript actually is able to explain at least some aspects of the experiment by Choi et al., so judgement on the issue may still be pending.

In addition, the authors should edit the abstract and introduction to their manuscript and remove formulations such as “The developed formalism reveals very general and….”. Furthermore, it seems to me the bibliography is somewhat biased towards self-citations of the authors’ works, and should be more equlibrated in this respect.

Requested changes

soften the claim that the model can be ruled out to explain the experiment by Choi et al.
edit abstract and introduction
equilibrate the bibliography with regards to citing own work and that of others

---

## Editorial Decision

published